# Sociodemographic Predictors of Changes in Physical Activity, Screen Time, and Sleep among Toddlers and Preschoolers in Chile during the COVID-19 Pandemic

**DOI:** 10.3390/ijerph18010176

**Published:** 2020-12-29

**Authors:** Nicolas Aguilar-Farias, Marcelo Toledo-Vargas, Sebastian Miranda-Marquez, Andrea Cortinez-O’Ryan, Carlos Cristi-Montero, Fernando Rodriguez-Rodriguez, Pia Martino-Fuentealba, Anthony D. Okely, Borja del Pozo Cruz

**Affiliations:** 1Department of Physical Education, Sports and Recreation, Universidad de La Frontera, Temuco 4780000, Chile; marcelo.toledo@ufrontera.cl (M.T.-V.); sebastian.miranda@ufrontera.cl (S.M.-M.); andrea.cortinez@ufrontera.cl (A.C.-O.); pia.martino@ufrontera.cl (P.M.-F.); 2UFRO Activate Research Group, Universidad de La Frontera, Temuco 4780000, Chile; 3IRyS Group, Physical Education School, Pontificia Universidad Católica de Valparaíso, Valparaíso 2374631, Chile; carlos.cristi@pucv.cl (C.C.-M.); fernando.rodriguez@pucv.cl (F.R.-R.); 4Early Start, Faculty of Arts, Social Sciences and Humanities, University of Wollongong, Wollongong, NSW 2522, Australia; tokely@uow.edu.au; 5Illawarra Health & Medical Research Institute, Wollongong, NSW 2522, Australia; 6Faculty of Health Sciences, Institute for Positive Psychology and Education, Australian Catholic University, Sydney, NSW 2060, Australia; borja.delpozocruz@acu.edu.au

**Keywords:** physical activity, sedentary behavior, sleep, active play, outdoor time, movement behaviors, COVID-19

## Abstract

The aim was to examine the sociodemographic predictors associated with changes in movement behaviors (physical activity, screen time, and sleep) among toddlers and preschoolers during the early stages of the coronavirus disease 2019 pandemic in Chile. Caregivers of 1- to 5-year-old children completed an online survey between 30 March and 27 April 2020. Information about the child’s movement behaviors before (retrospectively) and during the pandemic, as well as family characteristics were reported. In total, 3157 participants provided complete data (mean children age: 3.1 ± 1.38 years). During early stages of the pandemic, time spent in physical activity decreased, recreational screen time and sleep duration increased, and sleep quality declined. Toddlers and preschoolers with space to play at home and living in rural areas experienced an attenuated impact of the pandemic restrictions on their physical activity levels, screen time, and sleep quality. Older children, those whose caregivers were aged ≥35–<45 years and had a higher educational level, and those living in apartments had greater changes, mainly a decrease in total physical activity and increase in screen time. This study has shown the significant impact of the pandemic restrictions on movement behaviors in toddlers and preschoolers in Chile.

## 1. Introduction

The coronavirus disease 2019 (COVID-19) pandemic has had a significant impact on everyday life worldwide. Since the declaration of the global pandemic and the first national cases, Chile has had a total of 462,991 infected and suffered 12,741 deaths (from March through September 2020) [1]. During the pandemic, Chile has been in the top 10 countries with the highest ratio of deaths per 100 k population (67.9 deaths/100 k inhabitants by 30 September 2020) [2].

In response to this pandemic, the Chilean government implemented several approaches to reduce the health and economic impact of COVID-19. On 16 March 2020, all schools were closed, and, in the same week, workplaces implemented work-from-home strategies. On 17 March, all national parks were closed, and on 26 March some districts enforced lockdowns and curfews [3]. These restrictions likely had a negative impact on young children’s movement behaviors, (physical activity, sedentary behavior—including screen time, and sleep) [4,5]. Global guidelines recommend that children aged 1 to 5 should accumulate at least 180 min of physical activity, engage in no more than 1 h sedentary screen time (none for 1-year-olds), and have between 11–14 h (1- to 2-year-olds) and 10 to 13 h (3- to 5-year-olds) good-quality sleep per night [6,7]. Complying with these movement behavior guidelines, in one or more of several possible combinations of these recommendations (e.g., physical activity and sleeping) has been favorably associated with motor development, cognition, fitness, emotional regulation, adiposity, among others [6]. Limiting young children’s ability to meet these recommendations may have long-term health consequences [4,8]. Studies conducted during the COVID-19 pandemic have shown that children and adolescents have decreased physical activity, increased screen time, and slightly increased sleep duration [5]. However, there is a lack of data regarding the impact of COVID-19 on movement behaviors in children under 5 years of age and from Latin-American countries.

Understanding the association between COVID-19 factors and young children’s movement behaviors is important in guiding decision-making among policymakers, and in educating parents and health professionals. This information would be relevant not only during the pandemic but also when returning to a “new normal”. The purpose of this study was to examine the early impact of COVID-19 restrictions implemented in Chile on levels of physical activity, screen time, and sleep among toddlers and preschoolers. In particular, we aimed to explore the sociodemographic factors associated with changes in these movement behaviors in this population group.

## 2. Materials and Methods

Main caregivers of 1- to 5-year-old children living in Chile were invited to participate in an online survey from 30 March to 27 April 2020. The study was promoted through social media (Facebook, Twitter, and Instagram) and emails from educational institutions. Potential participants accessed a personalized link to read more detailed information about the study and gave their online informed consent to participate in the study. The inclusion criteria were: (1) living in Chile, (2) being the main caregiver of a 1- to 5-year-old child, and (3) living with the child most of the time before and during the COVID-19 pandemic. The study was approved by the Scientific Ethics Committee at Universidad de La Frontera, Chile (ORD.: 009-2020). 

Data were collected and managed using REDCap (Research Electronic Data Capture) [9] hosted at Universidad de La Frontera. Data collection started two weeks after the government enforced educational centers to close in Chile (16 March 2020) due to COVID-19. At the end of the data collection, the education centers were still closed at a national level (27 April 2020). 

### 2.1. Sociodemographic Variables

The online survey included questions about the caregiver, family, and the child participating in the study. The sociodemographic and home characteristics section of the survey included questions about educational level, family income, occupational situation, dwelling type and size, space to play at home, inhabitants per home, area of residence (urban/rural), city, and region. If there was more than one child at home between 1 to 5 years of age, the caregiver was asked to answer for only one of the children (freely chosen). Also, caregivers were asked about child sex, age, disability (yes/no and type), and enrolment in an early childhood education center (ECEC; yes/no). Caregivers were also asked if they were in lockdown at home at the moment of the survey, and the number of days spent in lockdown.

### 2.2. Movement Behaviors

A proxy-report from the main caregiver was used to assess children’s movement behaviors (physical activity, screen time, and sleeping). Caregivers were asked to provide the time (in hours and minutes) spent by the child on each of the movement behaviors before and during the COVID-19 pandemic. Sleep quality was assessed with a Likert scale (1 to 7; a higher score indicated better quality). The survey was piloted in a small sample before the official launch to optimize readability, length, and pertinence of the questions. The questions regarding movement behaviors are included as a Appendix A. The questions included in the study were based on those included in the International Study of Movement Behaviors in the Early Years (SUNRISE, www.sunrise-study.com) [10,11], but adaptations were made to capture the potential changes during the COVID-19 pandemic. Unpublished data from pilot studies in several countries assessed the concurrent validity of the questions for measuring time spent in total physical activity (TPA) and moderate-to-vigorous physical activity (MVPA) compared with ActiGraph GT3x accelerometers (ActiGraph, Pensacola, FL, USA) have shown significant but low correlations with both TPA (r = 0.14; *p* = 0.003) and MVPA (r = 0.16; *p* = 0.002), which is comparable with other proxy-report tools in this age group [12].

### 2.3. Statistical Analysis

Only participants with complete data for all sections were included in the analysis. Mean (standard deviation, SD), median (interquartile range, IQR), and proportions were used to describe participants on key characteristics and outcomes based on data distribution. Comparisons between sociodemographic characteristics were performed using parametric and non-parametric tests to compare behaviors before and during COVID-19. Multicollinearity was assessed with variable inflation factors (VIF). We used multiple linear regressions with a residualized change score approach [13,14] to investigate the sociodemographic predictors (independent variables) of changes in physical activity, recreational screen time, sleep duration, and sleep quality (dependent variables) during the early stages of the pandemic in Chile. This approach provides robust estimates by eliminating auto-correlated errors and regression towards the mean, which often makes it preferable to the simple change score approach [13]. First, we regressed the standardized score during the COVID-19 pandemic on the standardized scores before the COVID-19 pandemic for each of the behaviors assessed in this study. The residualized change score (i.e., trend) for each behavior was then estimated as the average of each participant’s residual score (i.e., the difference between the estimated value and the observed value). A positive residualized change score indicates an increase in the specific behavior from the time before COVID-19 and a negative score indicates a decrease. For each residualized change score, we explored the predictive role of a series of sociodemographic factors a priori thought to have influenced changes in the behaviors herein assessed after ECEC and school closures due to COVID-19. We adjusted all models for age and sex of the child, family income, presence of lockdown, and region. Data preparation and validation was conducted with Stata 15.0 (College Station, TX, USA: StataCorp LLC), while analyses were conducted in R (version 3.5.2) (R Foundation for Statistical Computing, Vienna, Austria). The level of significance was set at usual *p* < 0.05, two-tailed. 

## 3. Results

In total, 5505 individuals accessed the survey and 5266 accepted participation and signed the consent form. We had participants from each of the 16 regions in Chile. A total of 2109 (40.0%) participants were excluded from the analysis because they provided incomplete data (726 in demographics and COVID-19 related questions and 1383 for the movement behaviors). The final sample consisted of 3157 participants with complete data. Participants that provided complete data were more likely to be older (31.4 [6.06] vs. 30.3 [6.26], *p* < 0.001) and more educated (52.4% vs. 40.2% with university studies or more, *p* < 0.001) than those with incomplete data. The caregivers’ characteristics from those with complete data were comparable with those observed for the respective age group in the last National Census in terms of dwelling (81.6% vs. 79.7% living in a house) and living area (11.4% vs. 13.5% living in a rural area), but the current sample was more educated (65.7% vs. 39.7% with more than 12 years of education) [15]. The mean age of children was 3.1 (1.38) years. Most main caregivers were women (96.6%). Children enrolled in ECEC or schools were 28.1% of 1- to 2-year-olds and 71.9% of 3- to 5-year-olds. Most participants lived in houses (81.6%). About 9 out of 10 caregivers reported space to play at home. A large proportion of participants resided in urban areas (88.6%), and 76.3% s were under lockdown at the moment of the survey (mean duration 23.5 days [7.26]). Most sociodemographic characteristics were similar between boys and girls, except for the sex of the main caregiver (95.4% vs. 97.8%, *p* < 0.001), the number of people (4.2 vs. 4.1, *p* = 0.013) and children (1.6 vs. 1.7, *p* = 0.004) per home (Table 1).

### 3.1. Movement Behaviors before and during the Pandemic

Table 2 shows the changes in movement behaviors from before to during the COVID-19 pandemic by sex and age. Across all ages, mean time spent in physical activity decreased (−0.75 [CI 95% −0.81, −0.70] h/day), recreational screen time and sleep duration increased (1.4 [CI 95% 1.34, 1.45] h/day and 0.09 [CI 95% 0.04, 0.15] h/night, respectively) and sleep quality declined (−0.75 [CI 95% −0.81, −0.69] points). No differences by sex were observed for any of the movement behaviors before and during COVID-19. Older children were consistently less physically active, spent more time in screens, slept less, and had better sleep quality before and during the pandemic. More details according to other sociodemographic factors are shown in Appendix A.

### 3.2. Predictors of Change in Movement Behaviors

Table 3 shows the sociodemographic predictors of changes in movement behaviors due to the COVID-19 pandemic. Older children had greater reductions in physical activity and sleep duration, while they had more marked increase in their screen time. Children whose main caregiver was a man had smaller decline in physical activity and sleep quality. Those with caregivers aged ≥35–<45 years had greater reduction in physical activity and increase in screen time, while those with caregivers aged 45 and older had greater reduction in sleep duration. Children whose family had higher income had smaller decrease in sleep quality. Children whose main caregiver was more educated and those previously enrolled in an ECEC had greater reductions in their physical activity levels, and a greater increase in screen time. Toddlers and preschoolers living with five or more people had less marked decline in their physical activity, while those living with four or people had less marked increase in screen time. Children living with four people had greater reduction in sleep duration. Those living with more children had a less marked decline in their physical activity. Children who lived in apartment had greater reductions in physical activity and sleep quality, while they had a greater increase in screen time. Those living in a home different to a house or apartment had less marked increase in their screen time. Children who lived in homes with more square meters per person had a smaller decline in sleep quality. Children who had a space to play at home and those living in rural areas had smaller decline in physical activity and sleep quality, and less marked increase in screen time compared with their peers.

## 4. Discussion

To our knowledge, this is the first study examining the effects of COVID-19 on movement behaviors among children aged 1 to 5 years. We found that all movement behaviors changed across all ages, reflecting important secondary detrimental effects from this pandemic. The most common sociodemographic predictors of movement behaviors change during the COVID-19 pandemic were age, main caregiver’s sex, age, and education, family income, previous enrolment in an ECEC, dwelling type, available space to play, and type of residence area (urban/rural).

This is also the first known study in Chile that examined all three components of the 24-h movement behaviors in toddlers and preschoolers. A novel finding was that children from poorer families were more physically active than those in more affluent or educated families during the pandemic. This finding is different from that commonly observed in Chilean children and adolescents [16,17]. It is plausible that those caregivers were less likely to work from home under lockdown and kept more “normal” routines with their families, while those children may have had more freedom to play indoors and less demand to stay quiet during working hours. Also, having space to play at home was consistently related to healthier levels in all movement behaviors both before and during the pandemic. This finding reinforces the need for ensuring spaces for children at home and surrounding areas to play as this is likely to promote these behaviors not only during the pandemic but also in the return to a “new normal”. In Chile, specific content was developed for promoting physical activity through social media and national TV (since the 27 April 2020) as a response to the pandemic [18,19], and an expert committee on physical activity was installed by the Ministries of Sports, Health, and Education to inform, adapt and add physical activity content into the curriculum of the classes delivered remotely in this context [20]. Nevertheless, the lockdown measures did not allow people to participate in physical activity outdoors and most public parks were still closed during the period included in our study [3], with few councils prioritizing pedestrian zones to favor physical distancing [21]. These opposing strategies where on one hand physical activity is promoted indoors while at the same time it is severely restricted outdoors need to be periodically revised and aligned with the latest evidence, so health impairing over-restrictions are prevented. 

Preschoolers were the most affected by the restrictions during the pandemic, particularly for physical activity and recreational screen time. This may be explained by preschoolers needing more space to play and having greater access to screen-based devices than toddlers. The reduction in physical activity and the increase in screen time have particularly affected those with previous enrolment in an ECEC. Schools would be in a better position to respond with specific actions to prevent physical inactivity during this period than ECECs, as children and adolescents are more likely to be connected to participate in their virtual school activities than toddlers and preschoolers. Therefore, governments, policymakers, and professionals should devote particular attention to this age group. 

Our results suggest that more educated parents tend to restrict their children’s physical activity more, while at the same time may provide more opportunities to engage in screen-based behaviors. This is an interesting finding as, before the pandemic, children with more educated caregivers tended to engage less with screen-based devices than children with less-educated parents. This may be partially explained as more educated caregivers may have to work from home, and this, in turn, may require the caregiver to use screens to entertain their child while working from home. This may be even more complicated in families living in apartments, one of the strongest predictors of declines in physical activity and increases of screen time observed in this study during the pandemic. This finding has implications for urban planning. As the full impact of COVID-19 is uncertain and other pandemics may occur, upgrading substandard areas that are currently plenty of small social apartments with insufficient recreational public spaces should be considered, together with stronger regulation, as part of a plan to build healthier and more resilient cities. Undoubtedly, families and caregivers play a key role in facilitating movement behaviors of toddlers and preschoolers, but some political actions such as allowing specific time for those residing in apartments to go outside with their family may boost the process. Some caregivers and family characteristics (caregiver’s sex and age, number of people and children per home) were also associated with changes in movement behaviors during the pandemic, so strategies should consider messages and actions for the entire family group, particularly small families and those whose main caregivers are aged 35 and older [22].

Overall, sleep quality worsened, but children with higher family income and less crowded homes showed less marked declines. This aspect may reflect other social issues that are not captured in this study that are related to the economic situation in Chile [23]. The decrease in sleep quality has also been described in a study conducted in Italy in children aged 3 to 6 during the early stages of the pandemic; however, after two weeks of follow-up, this decline plateaued [24]. Some key messages to promote healthy sleep duration and quality, such as creating a bedtime routine and avoiding screens before sleep should be regularly disseminated and reinforced; and particularly tailored to low-income families [4,6,25].

Having space to play at home was the most consistent factor in the home environment predicting changes in movement behaviors. A novel finding from our study is that the play space was not only important for physical activity and screen time but also had an impact on sleep quality. This is particularly important as this association reinforce the call from the World Health Organization for focusing on the interaction of all three behaviors as they benefit each other [6]. Therefore, strategies originally thought for promoting active playing at home or surroundings may impact sleep, and this, in turn, may benefit the whole home environment. Another relevant factor that resulted in healthier changes in physical activity and screen time was living in a rural area. Recent studies conducted during the COVID-19 pandemic in children and adolescents have also reported that rurality is an important correlate of movement behaviors [26,27]. A study conducted in Croatia reported that the decrease in physical activity was more evident in adolescents living in urban areas than it was in those from rural areas [26]. For children living in Canada, factors such as living in an apartment and the proximity to major roads were barriers to engage in outdoor activities [27]. Also, adolescents spent more time outdoor if they lived in a low-density area and had access to parks in high-density neighborhoods [28]. We acknowledge that one of the primary measures taken by countries to control the pandemic was the imposition of mobility restrictions. However, decision-makers should seek options to facilitate outdoor recreational activities for the population while preserving safety and physical distancing instructions. Outdoor time is not only relevant for all movement behaviors but also the lack of it may impact other areas such as mental health, vitamin D deficiency and myopia [4]. In our study, the region of residence and lockdown situation were not predictors of the movement behavior changes suggesting that regardless of location and containment measures, movement behaviors, in both toddlers and preschoolers, were similarly affected during the pandemic in Chile.

The provision of safe spaces for physical activity is critical when physical distancing is required. As parks may be limited in space, availability and accessibility, programs such as open streets (e.g., Ciclovía) or play streets can be adapted and implemented while maintaining the physical distancing and other COVID-19-safe measures [29,30,31]. Thus, councils could provide additional safe spaces by closing lanes to cars, implementing complete streets schemes, and transforming them into spaces for physical activity, play, and recreation. Colombia has been highly innovative during the pandemic as they modified their current programs, usually delivered in parks or closed areas to face the needs of the population during the pandemic [32]. For example, in Bogotá, one of the first measures taken during the pandemic was the closure of lanes to private cars and the instalment of daily open streets [33]. Also, professionals from the Ministry of Sports Colombia have delivered their physical activity programs in the neighborhoods while maintaining an appropriate physical distance [32]. These low-cost initiatives have been positively received by the general public and policymakers [34,35], so once adapted to local needs they could be implemented in other contexts and countries, including Chile, and could help to alleviate the inactivity crisis [3]. 

The detrimental effects of the COVID-19 pandemic in toddlers and preschoolers are still uncertain [8]. If sustained, the COVID-19 lockdown measures may have a serious impact on children, with a greater effect on those from a more deprived background, increasing the gap in other developmental outcomes such as motor development, cognition, and literacy. Actions to prevent the adverse effects of the restriction measures due to COVID-19 are needed. As younger children are mostly dependent on adults, we should embrace this challenge and offer the best possible opportunities to promote a comprehensive development in the current context [8]. Messages and strategies should be respectful and appropriate for the families as many are suffering not only from the direct effects of the COVID-19 but also from economic and social hardships. These challenges are novel and highly demanding for governments and can only be successfully addressed if coordinated actions across sectors are undertaken. Cross-country collaborative efforts are needed to understand further how the pandemic is affecting movement behaviors of people from other ages, locations, and social groups. This may guide the allocation of resources where it is more critical. This study adds evidence that allows more balanced decision-making processes where not only the need to impose mobility restrictions to prevent contagion is taken into consideration but so is the need to avoid health impairments derived from extended confinements. Government departments are encouraged to develop and update their protocols in awareness of the multiple health impacts derived from the confinements.

### Strengths and Limitations

Strengths of this study include a large sample of participants from each of the Chilean regions. As recruitment was conducted online, voluntary response bias may have affected the sample composition with participants being more concerned regarding their children’s health and having therefore higher chances to notice impairments in their movement behaviors. Besides this limitation, the respondents’ characteristics were comparable from those observed in the last National Census for the respective age group in terms of dwelling type and living area, but the sample in our study was more educated [15]. Although our instrument was piloted before it was officially launched, we had a large percentage of incomplete questionnaires (40.0%). This could be explained since some people may be more reluctant to provide personal information through online platforms than in face-to-face modes. Some of the factors associated with the movement behaviors showed small predictive power (e.g., number of people at home). This may be explained as there are some other factors explaining these relationships. We recognize that movement behaviors are complex and depend on several dynamic factors that our questions were unable to capture, particularly during early stages of a pandemic. However, we measured these factors using the best assessment option possible under the very restricted circumstances imposed at the very early stages of the pandemic, when the data collection was conducted in Chile. We acknowledge that self-report measures used in this study may have biased the results (e.g., social desirability and recall). The use of accelerometers was not possible as physical distancing was mandated at the national level, including for those studies that were not strictly related to COVID-19 in clinical settings during the early stages of the pandemic [36]. Considering this, we collected data only for four weeks to minimize the risk of recall bias regarding the retrospective nature of some assessments. 

## 5. Conclusions

The COVID-19 lockdown measures have shown to have a serious impact on children’s behaviors. This study has shown that all movement behaviors changed during the early stages of the COVID-19 pandemic in toddlers and preschoolers in Chile. The most common sociodemographic factors associated with these changes were child’s age, main caregiver’s age, sex, and education, family income, main caregiver’s education, previous enrolment in an ECEC, dwelling type, space to play at home and type of residence area. Toddlers and preschoolers with space to play at home and living in rural areas had less marked impacts on physical activity, screen time, and sleep quality due to the pandemic. In contrast, older children, those whose caregivers were aged ≥35–<45 years and had higher educational level, and children living in apartments had greater changes, decreasing the TPA and increasing screen time. Together, the information provided in this study may help professionals and decision-makers to balance more precisely the health risks and benefits of confinements, inform future strategies and focus resources to reduce the potential adverse effects of the pandemic, immediately, and in the long-term.

## Figures and Tables

**Table 1 ijerph-18-00176-t001:** Sample characteristics.

	Total (*n* = 3157)	Boys (*n* = 1597)	Girls (*n* = 1560)	*p* ^a^
**Children’s age, mean years (SD)**	3.1 (1.38)	3.1 (1.38)	3.1 (1.38)	0.459
**Children’s age, *n*(%)**				
1–2 y	1243 (39.4)	635 (39.8)	608 (39.0)	0.833
3–4 y	1345 (42.6)	672 (42.1)	673 (43.1)	
5 y	569 (18.0)	290 (18.1)	279 (17.9)	
**Main caregiver’s sex, woman, *n* (%)**	3048 (96.6)	1523 (95.4)	1525 (97.8)	<0.001
**Main caregiver’s age, mean years (SD)**	31.4 (6.06)	31.2 (6.01)	31.6 (6.10)	0.098
**Main caregiver’s age category, *n* (%)**				
≥18–<25 years	440 (13.9)	236 (14.8)	204 (13.1)	0.076
≥25–<35 years	1769 (56.0)	888 (55.6)	881 (56.5)	
≥35–<45 years	894 (28.3)	454 (28.4)	440 (28.2)	
≥45 years	54 (1.7)	19 (1.2)	35 (2.2)	
**Family’s monthly income, *n* (%)**				
<530 USD	1201 (32.3)	504 (31.6)	517 (33.1)	0.589
≥530–<1830 USD	1546 (49.0)	787 (49.3)	759 (48.7)	
≥1830 USD	590 (18.7)	300 (19.1)	284 (18.2)	
**Main caregiver’s level of education, *n* (%)**				
Incomplete high school	102 (3.2)	49 (3.1)	53 (3.4)	0.716
Complete high school	982 (31.1)	499 (31.2)	483 (31.0)	
Technical degree	420 (13.3)	222 (13.9)	198 (12.7)	
University degree	1653 (52.4)	827 (51.8)	826 (52.9)	
**Children enrolled in ECEC, yes, *n* (%)**				
1- to 2-year-old	699 (28.1)	366 (29.1)	333 (27.1)	0.247
3- to 5-year-old	1786 (71.9)	891 (70.9)	895 (72.9)	
**Dwelling type, *n* (%)**				
House	2576 (81.6)	1310 (82.0)	1266 (81.1)	0.696
Apartment	475 (15.0)	232 (14.5)	243 (15.6)	
Other	106 (3.4)	55 (3.4)	51 (3.3)	
**Home size, *n* (%)**				
<50 m^2^	810 (25.7)	424 (26.6)	386 (24.7)	0.458
50 to <100 m^2^	1607 (50.9)	798 (50.0)	809 (51.9)	
≥100 m^2^	740 (23.4)	375 (23.5)	365 (23.4)	
**Number of people per home, mean (SD)**	4.1 (1.40)	4.2 (1.48)	4.1 (1.31)	0.013
**Number of children per home, mean (SD)**	1.7 (0.82)	1.6 (0.76)	1.7 (0.88)	0.004
**Children per home, *n* (%)**				
1 child	1511 (47.9)	778 (48.7)	733 (47.0)	0.026
2 children	1190 (37.7)	615 (38.5)	575 (36.9)	
3 or more	456 (14.4)	204 (12.8)	252 (16.1)	
**Squared meters per person at home, *n* (%)**				
<11.7 m^2^ per person	802 (25.4)	393 (24.6)	409 (26.2)	0.530
≥11.7 to <18.3 m^2^ per person	724 (22.9)	373 (23.4)	351 (22.5)	
≥18.3 to <25 m^2^ per person	721 (22.8)	357 (22.3)	364 (23.3)	
≥25 m^2^ per person	910 (28.8)	474 (29.7)	436 (28.0)	
**Available space to play, *n* (%)**				
Yes	2926 (92.7)	1481 (92.7)	1445 (92.6)	0.907
**Living area, *n* (%)**				
Urban	2797 (88.6)	1410 (88.3)	1387 (88.9)	0.584
Rural	360 (11.4)	187 (11.7)	173 (11.1)	
**Lockdown, *n*(%)**				
Yes	2409 (76.3)	1211 (75.8)	1198 (76.8)	0.524
**Number of days, mean (SD)**	23.5 (7.26)	23.267 (7.37)	23.2 (7.15)	0.786

Abbreviations: ECEC, early childhood care and education centers; USD, United States dollar. ^a^: significance when comparing characteristics between boys and girls.

**Table 2 ijerph-18-00176-t002:** Description of movement behaviors before and during the early stages of the COVID-19 pandemic in toddlers and preschoolers in Chile according to age and sex.

	Physical Activity Mean (SD), h/Day			Screen Time Mean (SD), h/Day			Sleep Duration Mean (SD), h/Day			Sleep Quality Mean (SD), Score 1 to 7		
**Characteristic**	Before COVID-19	During COVID-19	*p*	Before COVID-19	During COVID-19	*p*	Before COVID-19	During COVID-19	*p*	Before COVID-19	During COVID-19	*p*
**Total**	3.6 (1.97)	2.82 (2.15)	<0.001	1.66 (1.15)	3.05 (1.92)	<0.001	10.92 (1.80)	11.01 (1.86)	0.001	5.68 (1.54)	4.93 (1.77)	<0.001
**Sex**												
Girls	3.6 (2.00)	2.8 (2.14)	<0.001	1.65 (1.17)	3.00 (1.86)	<0.001	10.86 (1.75)	10.99 (1.82)	0.001	5.67 (1.56)	4.92 (1.79)	<0.001
Boys	3.6 (1.94)	2.9 (2.15)	<0.001	1.66 (1.13)	3.09 (1.98)	<0.001	10.97 (1.84)	11.02 (1.90)	0.214	5.69 (1.52)	4.93 (1.76)	<0.001
**Age in years**												
1	3.94 ^a^ (2.17)	3.65 ^b^ (2.42)	<0.001	1.33 ^a^ (1.03)	2.14 ^b^ (1.66)	<0.001	12.11 ^a^ (1.95)	11.96 ^b^ (2.03)	0.024	5.26 ^a^ (1.58)	4.58 ^b^ (1.76)	<0.001
2	3.92 (2.18)	3.14 (2.29)	<0.001	1.56 (1.12)	2.77 (1.73)	<0.001	11.35 (1.72)	11.38 (1.92)	0.710	5.54 (1.53)	4.83 (1.79)	<0.001
3	3.59 (1.96)	2.70 (2.01)	<0.001	1.77 (1.20)	3.29 (1.91)	<0.001	10.79 (1.61)	10.82 (1.70)	0.645	5.68 (1.57)	5.05 (1.69)	<0.001
4	3.22 (1.75)	2.42 (1.87)	<0.001	1.76 (1.13)	3.33 (1.98)	<0.001	10.33 (1.59)	10.57 (1.71)	<0.001	5.89 (1.49)	5.04 (1.81)	<0.001
5	3.23 (1.62)	2.25 (1.81)	<0.001	1.84 (1.21)	3.66 (1.97)	<0.001	10.07 (1.33)	10.38 (1.47)	<0.001	6.01 (1.42)	5.12 (1.76)	<0.001

Abbreviations: COVID-19, coronavirus disease 2019; ^a^: *p*-value < 0.05 when comparing each behavior between categories before COVID-19. ^b^: *p*-value < 0.05 when comparing each behavior between categories during COVID-19.

**Table 3 ijerph-18-00176-t003:** Sociodemographic predictors of change in movement behaviors in toddlers and preschoolers during COVID-19 pandemic in Chile.

Characteristic	Residualized Change Score in Physical Activity (SD) (*n* = 3157)	Residualized Change Score in Screen Time (SD) (*n* = 3157)	Residualized Change Score in Sleep Duration (SD) (*n* = 3157)	Residualized Change Score in Sleep Quality (SD) (*n* = 3045)
**Child’s sex (ref: girls)**	0.033 (0.035)	0.056 (0.035)	−0.032 (0.036)	−0.003 (0.036)
**Child’s age, years (ref: 1 year)**	−0.121 *** (0.013)	0.155 *** (0.013)	−0.047 *** (0.013)	0.019 (0.013)
**Caregiver’s sex (ref: woman)**	0.201 * (0.098)	−0.169 (0.098)	−0.058 (0.098)	0.252 * (0.099)
**Caregiver’s age (ref: 18–<25 years)**				
≥25–<35 years	−0.104 (0.055)	0.096 (0.055)	−0.077 (0.055)	0.064 (0.056)
≥35–<45 years	−0.244 *** (0.062)	0.216 ***(0.062)	−0.113 (0.062)	0.093 (0.063)
≥45 years	−0.229 (0.145)	−0.127 (0.145)	−0.387 *** (0.145)	0.162 (0.146)
**Family income per month** **(ref: <530 USD)**				
≥530–<1830 USD	−0.058 (0.040)	0.056 (0.039)	−0.016 (0.040)	0.109 ** (0.041)
≥1830 USD	−0.062 (0.040)	0.051 (0.051)	−0.010 (0.052)	0.294 *** (0.053)
**Main caregiver’s level of education** **(ref: Incomplete high school)**				
Complete high school	−0.182 (0.102)	0.177 (0.102)	0.013 (0.104)	0.121 (0.107)
Technical degree	−0.300 ** (0.109)	0.282 ** (0.108)	−0.099 (0.111)	0.104 (0.114)
University degree	−0.371 *** (0.104)	0.271 ** (0.103)	−0.093 (0.106)	0.122 (0.109)
**Children enrolled in ECEC (ref: no)**				
Yes	−0.195 *** (0.048)	0.391 *** (0.048)	0.078 (0.049)	−0.032 (0.050)
**Dwelling type (ref: House)**				
Apartment	−0.404 *** (0.049)	0.226 *** (0.049)	−0.089 (0.051)	−0.109 * (0.051)
Other	0.114 (0.098)	−0.219 * (0.098)	0.030 (0.100)	0.031 (0.101)
**Number of people per home** **(ref: ≤3)**				
4	0.058 (0.042)	−0.088 * (0.042)	−0.085 * (0.043)	−0.009 (0.044)
≥5	0.115 ** (0.043)	−0.112 ** (0.043)	0.004 (0.044)	−0.049 (0.045)
**Children per home** **(ref: 1 child)**				
2 children	0.112 ** (0.038)	−0.021 (0.038)	−0.015 (0.039)	−0.023 (0.040)
3 or more	0.124 * (0.053)	−0.025 (0.053)	0.067 (0.054)	0.006 (0.055)
**Squared meters per person at home** **(ref: <11.7 m^2^ per person)**				
≥11.7–<18.3 m^2^ per person	−0.042 (0.052)	0.049 (0.051)	−0.074 (0.052)	0.051 (0.053)
≥18.3–<25 m^2^ per person	−0.038 (0.053)	0.079 (0.053)	−0.025 (0.054)	0.087 (0.055)
≥25 m^2^ per person	0.053 (0.053)	−0.007 (0.053)	0.025 (0.054)	0.158 ** (0.055)
**Available space to play (ref: no)**				
Yes	0.546 *** (0.067)	−0.398 *** (0.067)	−0.057 (0.069)	0.396 *** (0.070)
**Living area (ref: urban)**				
Rural	0.563 *** (0.055)	−0.278 *** (0.055)	0.102 (0.056)	0.120 * (0.057)
**Lockdown (ref: no)**	−0.074 (0.042)	0.050 (0.041)	−0.076 (0.042)	−0.076 (0.043)

Abbreviations: COVID-19, coronavirus disease 2019; ECEC, early childhood care and education centers. Models adjusted by child’s sex, age, family income, presence of lockdown, and region. Significance: *** = *p* < 0.001; ** = *p* < 0.01; * = *p* < 0.05.

## Data Availability

The data presented in this study can be shared based on specific request from the corresponding author.

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
