# Peer review of "Sociodemographic Predictors of Changes in Physical Activity, Screen Time, and Sleep among Toddlers and Preschoolers in Chile during the COVID-19 Pandemic"

_ijerph, 2020, doi:10.3390/ijerph18010176_

Round 1

Reviewer 1 Report

An interesting study that was done during the pandemic, highlighting its impact on children's movement.

Abstract

Well summarized the study

Introduction

Can be further strengthened on why we need to study about children's movement behaviours. 

Methods

Well described.

Results

Table 1 - provide numbers along with the % in brackets for categorical variables

Rest of the section is well presented. 

Discussion

Discussion is thoroughly done. 

Conclusion

Well summarized the findings. 

Author Response

Reviewer 1

An interesting study that was done during the pandemic, highlighting its impact on children's movement.

 Introduction

Can be further strengthened on why we need to study about children's movement behaviours. 

Re: We have added a sentence to briefly explain the benefits of meeting some of the recommendations. Then this sentence is connected to a previous one indicating that limiting the ability these recommendations may have long-term health consequences (see below).

 L55, Complying with these movement behaviour guidelines, in one or more of several possible combinations of these recommendations (e.g.., physical activity and sleeping) has been favourably associated with motor development, cognition, fitness, emotional regulation, adiposity, among others [6]. Limiting young children's ability to meet these recommendations may have long-term health consequences [4,8].

Results

Table 1 - provide numbers along with the % in brackets for categorical variables

Rest of the section is well presented. 

Re: we have added the % in brackets for categorical variables as suggested.

Reviewer 2 Report

Does the introduction provide sufficient background and include all relevant references?

The introduction is adequate and uses updated bibliographic references.

Is the research design appropriate?

Although the study design has several limitations, these have been included in the limitations section of the manuscript.

Are the methods adequately described?

Please include the reliability and validity of the questions used to assess Movement behaviours (physical activity, screen time, and sleeping).

Are the results clearly presented?

The results are clear and adequate.

Are the conclusions supported by the results?

The conclusions are according to the results obtained, they are correct.

In my opinion, the study has several limitations:

1. It is a convenience sample and not representative.
2.  There is a significant bias in conducting the survey via the Internet, given that not all families have access to the Internet.
3. 40% of the questionnaires were not fully completed.
4. The reliability and validity of the proxy report used to assess children's movement behaviour (physical activity, screen time, and sleeping) is not reported. This information is important for determining the validity of the data obtained in the study. Accelerometers would have been more appropriate.

Limitations 1, 2, and 3 are listed in the limitations section of the study. It is necessary to include the reliability and validity of the proxy report used to assess children's movement behaviour (physical activity, screen time, and sleeping).

The main strengths of the study are:

1. This is the first study examining the effects of COVID-19 on movement behaviours among children aged 1 to 5 years. However, it should be noted that this effect cannot be extrapolated to other countries, as the measures taken to control the pandemic were different in each country.

2. This is the first known study in Chile that examined all three components of the 24-hour movement behaviours in toddlers and pre-schoolers.

As for suggestions for improvement, I think it would be necessary to review formal aspects of the article such as placing the number of bibliographic citations before the points, this should be the other way around. I am surprised that you did not ask for the sex and age of the child's primary caregiver, was this information requested? if so, please include it. do you think it might have been relevant to include this information in your analyses?

In general, I think it is a pioneering study, which although it has several important limitations, have been reflected in the limitations section, so after including the recommendations made I think it could be published in this journal.

Author Response

Are the methods adequately described?

Please include the reliability and validity of the questions used to assess Movement behaviours (physical activity, screen time, and sleeping).

Re: we have added information regarding unpublished data about the reliability reports from the SUNRISE Study.

In my opinion, the study has several limitations:

1. It is a convenience sample and not representative.
2.  There is a significant bias in conducting the survey via the Internet, given that not all families have access to the Internet.
3. 40% of the questionnaires were not fully completed.
4. The reliability and validity of the proxy report used to assess children's movement behaviour (physical activity, screen time, and sleeping) is not reported. This information is important for determining the validity of the data obtained in the study. Accelerometers would have been more appropriate.

Limitations 1, 2, and 3 are listed in the limitations section of the study. It is necessary to include the reliability and validity of the proxy report used to assess children's movement behaviour (physical activity, screen time, and sleeping).

The main strengths of the study are:

1. This is the first study examining the effects of COVID-19 on movement behaviours among children aged 1 to 5 years. However, it should be noted that this effect cannot be extrapolated to other countries, as the measures taken to control the pandemic were different in each country.

Re: we have added these implications in the discussion, and we have highlighted that the message is particularly relevant for Chile. Nevertheless, we believe it also may be helpful in other contexts.

  1. This is the first known study in Chile that examined all three components of the 24-hour movement behaviours in toddlers and pre-schoolers.

As for suggestions for improvement, I think it would be necessary to review formal aspects of the article such as placing the number of bibliographic citations before the points, this should be the other way around.

Re: Thanks for noticing this. We have corrected this in the manuscript.

I am surprised that you did not ask for the sex and age of the child's primary caregiver, was this information requested? if so, please include it. do you think it might have been relevant to include this information in your analyses?

Re: We have included the main caregiver’s sex in table 1, but the caregiver’s age was already included. We have added the new analysis and presented the results in tables.

In general, I think it is a pioneering study, which although it has several important limitations, have been reflected in the limitations section, so after including the recommendations made I think it could be published in this journal.

Re: Thank you for your comments. We have included several changes to the manuscript based on your suggestions. We think they made our manuscript clearer.

Reviewer 3 Report

The article presents a current topic of interest to the scientific community and society, but for its publication, I consider that some of the aspects that I describe below should be completed and clarified.

1. In the method section, it must be expressed together with the measurement instrument used in the present study, its factors and the reliability analysis for each factor (through the cronbach alpha and omega statistic).

2. The authors have to indicate the regression analysis method used, as well as the collinearity statistics, and the independent and dependent variables used in the analysis prior to the inclusion of Table 3.

3. It should be justified in the discussion because there is little predictive power in many of the variables expressed in the analyzes.

Author Response

Reviewer 3

  1. In the method section, it must be expressed together with the measurement instrument used in the present study, its factors and the reliability analysis for each factor (through the cronbach alpha and omega statistic).

Re: we have added details obtained from unpublished data from the SUNRISE study (See references 10 and 11. The questions were based in the instrument used in that international study, but they were modified accordingly to capture potential changes in movement behaviours during the pandemic.

  1. The authors have to indicate the regression analysis method used, as well as the collinearity statistics, and the independent and dependent variables used in the analysis prior to the inclusion of Table 3.

Re: We have added the details requested in the methods section as suggested. Also, we have included a clarification in brackets in the same section to highlight the dependent and independent variables). Control variables used in the multiple linear regressions were already described in methods and below the table 3.

  1. It should be justified in the discussion because there is little predictive power in many of the variables expressed in the analyzes. (L367, P11)

Re: We have added some sentences in the strengths and limitations section of the manuscript to elaborate on this issue.

Reviewer 4 Report

Dear authors, it is a pleasure for me to review an article that, despite being one more that has emerged as a result of the coronavirus pandemic that has changed the world, provides information that had not been analysed until now.

It is a well-elaborated and structured article from the abstract to the conclusions, including a well structured and developed methodology section.

However, there are certain considerations regarding formal and presentation aspects of the manuscript that the authors should consider before granting my final acceptance of the manuscript.

First, citations in the text should be included before the endpoint of the sentence, not after it. Please review this aspect throughout the document.

Secondly, with respect to lines 143 and 144, authors should change the spacing after the table instead of leaving two lines blank.

With regard to the presentation of the results, I would recommend that the authors of the manuscript review and edit tables 1 and 3 to try to fit them on one page, not two pages. The reason for this suggestion is to improve the reading and understanding of them.

Finally, the authors should review and adjust the bibliographic references in the style recommended by the journal.

Author Response

However, there are certain considerations regarding formal and presentation aspects of the manuscript that the authors should consider before granting my final acceptance of the manuscript.

First, citations in the text should be included before the endpoint of the sentence, not after it. Please review this aspect throughout the document.

Re: we have formatted the manuscript accordingly.

Secondly, with respect to lines 143 and 144, authors should change the spacing after the table instead of leaving two lines blank.

Re: we have changed this detail.

With regard to the presentation of the results, I would recommend that the authors of the manuscript review and edit tables 1 and 3 to try to fit them on one page, not two pages. The reason for this suggestion is to improve the reading and understanding of them.

Re: We have changed the format. We are open for suggestions from the journal editorial team.

Finally, the authors should review and adjust the bibliographic references in the style recommended by the journal.

Re: we have used the endnote file recommended by the editorial.